# A New Thermal Elasto-Hydrodynamic Lubrication Solver Implementation in OpenFOAM

James Layton, Benjamin C. Rothwell * , Stephen Ambrose, Carol Eastwick, Humberto Medina and Neville Rebelo

Gas Turbine and Transmissions Research Center, Energy Technologies Building, University of Nottingham, Nottingham NG7 2TU, UK; eayjal@nottingham.ac.uk (J.L.); stephen.ambrose3@nottingham.ac.uk (S.A.); carol.eastwick@nottingham.ac.uk (C.E.); humberto.medina@nottingham.ac.uk (H.M.); neville.rebelo@nottingham.ac.uk (N.R.)
* Correspondence: benjamin.rothwell@nottingham.ac.uk

**Abstract:** Designing effective thermal management systems within transmission systems requires simulations to consider the contributions from phenomena such as hydrodynamic lubrication regions. Computational fluid dynamics (CFD) remains computationally expensive for practical cases of hydrodynamic lubrication while the thermo elasto-hydrodynamic lubrication (TEHL) theory has demonstrated good accuracy at a lower computational cost. To account for the effects of hydrodynamic lubrication in high-power transmission systems requires integrating TEHL into a CFD framework such that these methodologies can be interfaced. This study takes an initial step by developing a TEHL solver within OpenFOAM such that the program is prepared to be interfaced with a CFD module in future versions. The OpenFOAM solver includes the Elrod–Adams cavitation model, thermal effects, and elastic deformation of the surfaces, and considers mixing between the recirculating flow and oil feed by applying energy and mass continuity. A sensitivity study of the film mesh is presented to show the solution variation with refinement along the circumferential, axial and radial directions. A validation case is presented of an experimental single axial groove journal bearing which shows good agreement in the pressure and temperature results. The peak pressure in the film is predicted within 12% and the peak temperature in the bush is predicted within 5% when comparing the centerline profiles.

**Keywords:** hydrodynamic lubrication; journal bearing; TEHL; OpenFOAM





## 1. Introduction

Emission targets set for the European aircraft industry in Flightpath 2050 [1] include reducing $CO_2$ emissions by 75%, NOx emissions by 90%, and noise by 65%. One challenge of achieving these targets is improving energy efficiency in gas turbine engines, requiring higher operating temperatures and power densities which place a greater thermal load on the thermal management system while being restricted by weight and volume constraints. To maximise efficiency and improve thermal management designs requires thermal fluid simulations to accurately consider sources of heat in the system.

Hydrodynamic lubrication is a phenomenon where a wedge of fluid film is formed between sliding surfaces due to their relative motion. The effect appears in converging–diverging contact regions, for example, in journal bearings, ball bearing and gear teeth meshing. Under a full-film lubrication regime, there is no contact between the surfaces and the load is carried by the lubricating fluid; therefore, the effect is commonly employed in industry to maximise the lifetime of components, such as in journal bearings. High shearing and pressure gradients are imposed on the lubricating fluid as it passes between the surfaces, causing considerable heat generation as the operating speed and load is increased.

Developments in computer technology have allowed researchers to adopt computational fluid dynamics (CFD) approaches in investigations of hydrodynamic lubrication. The

thin geometries in hydrodynamic lubrication regions require a high mesh refinement and often simplified approaches are taken in CFD studies. Hartinger et al. [2] and Hajishafiee et al. [3] applied two-dimensional CFD methodologies to hydrodynamic lubrication regimes of line contacts where computational times were of the order of days. Dhande et al. [4] developed a three-dimensional multiphase CFD methodology for a journal bearing in isothermal conditions where the mesh required a radial discretisation of four elements and a total mesh size in the film of 102,636. Qiang Li et al. [5] developed a full 3D CFD-CHT methodology to investigate the three-dimensional temperature distribution in a journal bearing which required 1,159,000 elements in the film region. For industrial contexts, where two-dimensional approximations are inappropriate and deformation is significant, it becomes impractical to use CFD to simulate hydrodynamic lubrication regions because of the high computational requirements.

Hydrodynamic lubrication theory models the lubricating fluid within a converging–diverging gap at a low computational cost while maintaining good accuracy when compared to experimental results. The methodology is based on the thin-film assumptions from which the Reynolds Equation [6] is derived for describing the pressure in the lubricating fluid. The Reynolds Equation predicts sub-ambient pressures in the diverging region where, in practice, the fluid tends to cavitate. There are two mechanisms of cavitation in this region: gaseous cavitation, where dissolved gases are released from the fluid, and vapor cavitation, where the lubricating fluid vaporises when its pressure drops below its vapor pressure. Various approaches have been taken to account for the cavitation effects in the lubricant. Jakobsson and Floberg [7] used a switch function to differentiate the film and cavitation regions with Olsson [8] extending it to transient conditions. Elrod and Adam [9] developed an algorithm for cavitation which also uses a switch function with a pressure-density function. A single equation is applied to the domain which naturally locates the cavitation boundary and maintains mass continuity in its solution. However, the coefficient matrix formed when the equation is discretised tends to be stiff due to the high compressibility of the fluid, which leads to difficulties in reaching a converged solution. This problem is overcome by reducing the bulk modulus of the fluid, which assists numerical convergence [9]. Vijayaraghavan and Keith [10] extended the Elrod–Adams methodology to switch between upwind differencing in the cavitation region and central differencing in the full-film region, allowing a second-order solution in the full-film region. Fesanghary et al. [11] introduced a modification to the switch function algorithm to reduce oscillations and instability in the solution.

Other approaches to the cavitation problem in hydrodynamic lubrication include Bayada et al. [12] who used a pressure-density function with the bulk modulus based on the speed of sound in the fluid to encompass the cavitation effects. The results compared well against the Elrod–Adams model and without the requirement of artificially reducing the bulk modulus. Similarly, Ransegnola et al. [13] developed a fluid state model which included simulating both the release of dissolved gas and vaporisation of the fluid. These models have improved physical accuracy and the equations can be derived directly from the governing equations without applying methods such as the switch function or reducing the bulk modulus. However, these methods introduce additional non-linearity in the bulk modulus functions, which leads to further numerical instability and convergence issues.

The hydrodynamic lubrication theory has been extended to consider thermal effects due to the heat dissipated within the film. Olsson [8] and Alakhramsing et al. [14] used an adiabatic assumption where conduction to the solids was neglected and a two-dimensional form of the energy equation was derived. The maximum temperature in the solid is a notable cause of failure in mechanical components operating under hydrodynamic lubrication and, therefore, is a valuable quantity to predict. Stefani et al. [15] used a quasi-3D approach employing a parabolic temperature profile across the film. Studies such as Banwait and Chandrawat [16] solved the energy equation across the entire domain, including the temperature profile across the film height. Heat generated from shearing

of the fluid can be significant enough to vary the fluid properties, therefore, affecting the pressure and temperature solutions.

Surface deformation influences the film height within the contact region affecting the pressure and temperature solution. Lingamaa et al. [17] used a finite element approach to calculate the deformations within the solid regions. The half-space approximation [18] is often used to estimate the elastic deformation, removing the requirement for an FEM setup within the solid regions. Bouyer and Fillon [19] approximated the thermal deformation using the averaged temperature of the shaft surface and the average temperature in the bush.

The finite difference method (FDM) is commonly applied to discretise the TEHL equations, such as in Zhang et al. [20] where the lubrication of a journal bearing operating in a turbulent regime was modelled. The finite difference method requires an ordered mesh and is not practical for complex geometries. Giacopini et al. [21] took a finite-element-based approach for solving the complementary condition which forms due to cavitation in hydrodynamic lubrication. The finite element method (FEM) is effective for irregular geometries, although implementing the Elrod–Adams cavitation approach "is not straightforwardly accomplished" [22]. An element-based finite volume method (EbFVM) was described in Profito et al. [22] to enforce local and global continuity of the transport equations in the region, while maintaining the advantages of the FEM for irregular meshes. A finite area approach was taken in Skuric et al. [23] using OpenFOAM, which excluded deformation and thermal effects.

Accurate consideration of hydrodynamic lubrication effects are often excluded in simulations of high-power transmission systems. The hydrodynamic lubrication modelling theory based on the original work by Reynolds provides an accurate model with low computational requirement for this fluid regime. Many studies have extended the theory into a multi-physics model which includes thermal effects and surface deformation. The models generally only consider the lubrication domain and are not implemented in a framework to allow interfacing with CFD. In this paper, a TEHL solver is developed in OpenFOAM using a Reynolds-based methodology within a CFD framework which will allow for interfacing with a CFD module. The aim of this paper is to develop a Reynolds-based hydrodynamic lubrication model within a CFD framework which includes cavitation, thermal effects and surface deformation.

## 2. Materials and Methods

### 2.1. Thermo-Elastohydrodynamic Lubrication Model

#### 2.1.1. Reynolds Equation

The Reynolds Equation describes the pressure in a hydrodynamic lubrication regime. The compressible steady-state vector form of the equation can be written as

$$\nabla \cdot \left( \frac{\rho h^3}{12\mu} \nabla p \right) = \nabla \cdot (\vec{U}_c \rho h) \tag{1}$$

where $\rho$ is the density $\left( \frac{\text{kg}}{\text{m}^3} \right)$, $\mu$ is the dynamic viscosity (Pas), p is the pressure (Pa), h is the film height (m), and $\vec{U}_c$ is the Couette velocity $\left( \frac{\text{m}}{\text{s}} \right)$ approximated by the average velocity between the adjacent surfaces.

#### 2.1.2. Film Height

The film height is calculated from the sum of three components,

$$h = h_e + h_d + h_T \tag{2}$$

where $h_e$ is the film height profile generated by the eccentricity of the shaft and is calculated from the equation

$$h_e = c(1 - \epsilon cos(\theta)) \tag{3}$$

where c is the bearing clearance (m), $\epsilon$ is the eccentricity, and $\theta$ is the angular position (°). The surface deformation, $h_d$, is calculated using the half-space approximation [18], which takes the form of an integral equation,

$$h_d(x,z) = \frac{2\pi}{E'} \int \int \frac{p(x_i, z_k)}{\sqrt{(x - x_i)^2 + (z - z_k)^2}} dx_i dz_k \tag{4}$$

where $x$ and $z$ are the circumferential and axial positions (m), respectively, and $E'$ is the adjusted Young's modulus

$$\frac{2}{E'} = \frac{1 - v_1^2}{E_1} + \frac{1 - v_2^2}{E_2}. \tag{5}$$

where $E$ is the Young's modulus of the solid material (Pa) and $v$ is the Poisson's ratio. The thermal dilation, $h_T$, is calculated using a similar approach to Bouyer et al. [19]. The average temperature along the film–solid interface is used to estimate the change in radius of the surface. The shaft expansion decreases the film height while the bush expansion increases the film height; therefore, the total thermal dilation is calculated as

$$h_T = \alpha_b R_b (\bar{T}_b - T_{ref}) - \alpha_s R_s (\bar{T}_s - T_{ref}) \tag{6}$$

where $\alpha$ is the thermal expansion coefficient $(K^{-1})$, $R_b$ and $R_s$ are the radius of the bush and shaft, respectively (m), $\bar{T}_b$ and $\bar{T}_s$ are the average temperature of the bush and shaft (K), respectively, and $T_{ref}$ is the reference temperature (K). Given the expected magnitude of effect from the thermal dilation, which in Bouyer et al. [19] changed the minimum film thickness by 0.1 µm, this approximation is appropriate given the minimal computational requirement.

### 2.1.3. Temperature Equation

The steady-state energy equation is written as:

$$\nabla \cdot (c_p \rho \vec{U} T) - k \nabla^2 T = \mu \left( \frac{\vec{U}}{h} \right)^2 \tag{7}$$

where $c_p$ is the specific heat capacity $\left( \frac{J}{kg\,K} \right)$, $k$ is the thermal conductivity $\left( \frac{W}{Km} \right)$, and $\vec{U}$ is the fluid velocity calculated from the sum of the Couette and Poiseuille flows:

$$\vec{U} = \left( U_s \frac{y}{h} \right) \cdot \hat{x} - \left( \frac{y^2 - hy}{2\mu} \right) \nabla p \tag{8}$$

where $y$ is the radial position within the film (m), $\hat{x}$ is the unit vector tangential to the bush surface, and $U_s$ is the velocity of the shaft surface $\left( \frac{m}{s} \right)$.

### 2.1.4. Fluid Viscosity

The viscosity of the fluid is defined as a function of temperature using the power law model:

$$\mu = \mu_0 e^{\gamma(T - T_{ref})} \tag{9}$$

where $\mu_0$ is the viscosity at the reference temperature, $T_{ref}$, and $\gamma$ is the viscosity-temperature coefficient.

### 2.1.5. Cavitation

The Elrod–Adams cavitation algorithm is applied where a switch function is used to define a full-film region and a cavitation region. The relative density, $\Theta$, is introduced and related to the pressure of the fluid:

$$\Theta = \frac{\rho}{\rho_0} \tag{10}$$

$$P = P_{cav} + g\beta \ln(\Theta) \tag{11}$$

where $\beta$ is the compressibility of the fluid (Pa) and $g$ is the switch which enforces the constant pressure condition in the cavitation region. The values of $\Theta$ can be expected to be close to 1 as the value of $\beta$ is high for fluids; therefore, Equation (11) can then be approximated by

$$P = P_{cav} + g\beta(\Theta - 1) \tag{12}$$

The switch, g, is adjusted based on $\Theta$:

$$\begin{aligned} \Theta < 1, \quad g = 0 \\ \Theta \geq 1, \quad g = 1 \end{aligned} \tag{13}$$

Substituting Equation (11) into Equation (1) gives a single equation to be applied to the film domain:

$$\nabla \cdot \left( \frac{g\beta h^3}{12\mu} \nabla \Theta \right) = \nabla \cdot (\vec{U}\Theta h) \tag{14}$$

2.1.6. Material Properties

The lubricating fluid is modelled as a fluid–vapor mixture where material properties are approximated using the fluid volume fraction. Gaseous cavitation is assumed where the vapor phase is comprised of undissolved gases released from the lubricant. The gaseous phase is, therefore, assumed to be air, and the material properties are estimated using the equation:

$$\zeta = \omega\zeta_l + (1 - \omega)\zeta_g \tag{15}$$

where $\zeta$ is a fluid property which includes $\mu$, $\rho$, $c_p$ and $k$ [24]. The fluid volume fraction, $\omega$, is the value of $\Theta$ in the cavitation region and is 1 in the full-film region.

2.2. *Solid Region Models*

The temperature in the bush is modelled using the Laplacian equation:

$$k\nabla^2 T = 0 \tag{16}$$

For the shaft, a solid body convective term is included to account for the rotation of the shaft:

$$c_v\rho\nabla \cdot (\vec{U}T) - k\nabla^2 T = 0 \tag{17}$$

2.3. *Boundary Conditions*

Figure 1 shows the journal bearing geometry and the locations of the boundaries. The boundary condition types are summarised in Table 1. The pressure is calculated from the relative density using Equation (11); therefore, boundary conditions are applied to $\Theta$ in the TEHL procedure. At the boundaries to the solid bodies, a fixed gradient of zero is applied to the $\Theta$ field to enforce the thin-film assumption that there is negligible pressure variation across the film height. At the inlet and outlet of the film region, a fixed value is applied which corresponds to the supply pressure of the bearing, calculated using Equation (11). An atmospheric pressure condition is appropriate for the side boundaries of the pressure field within the film. As gaseous cavitation is assumed and cavitation pressure is equal to atmospheric pressure, then, at atmospheric pressure, $\Theta$ can take any value between 0 and 1, following from Equation (11). In the converging region, it is expected that the pressure will force the lubricant outwards as side leakage, and a fixed value of $\Theta = 1$ is appropriate. However, in the diverging region, pressure is at or below atmospheric pressure; therefore, no side leakage is expected and the boundary appears as an inlet. A fixed value of $\Theta = 1$

is not appropriate here as this assumes a supply of lubricant is available. To account for this, the side boundary condition varies between a fixed value of one and a fixed gradient of zero, depending on whether the boundary is adjacent to the full-film or the cavitation region, respectively. This boundary condition is updated by the switch function in the cavitation algorithm.

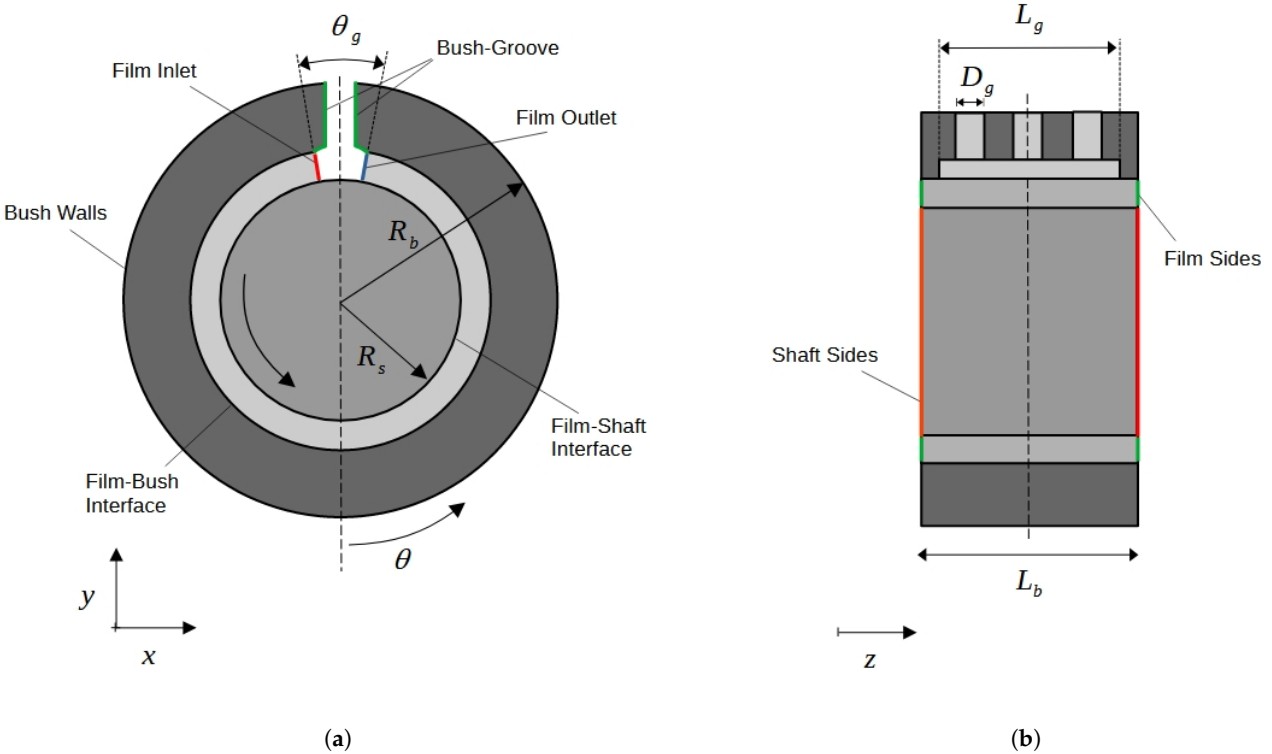

(a)            (b)

**Figure 1.** Journal bearing geometry and locations of the boundaries: (**a**) Front, (**b**) Side.

**Table 1.** Boundary Condition Types.

| Boundary Name | Boundary Type | |
| --- | --- | --- |
| | Relative Density, $\Theta$ (-) | Temperature, T ($^\circ$C) |
| Film Inlet | Dirichlet | Dirichlet |
| Film Outlet | Dirichlet | Neumann |
| Film Sides | Dirichlet/Neumann | Neumann |

At the film–solid interface, heat continuity is applied with the neighbouring solid region using a coupled boundary condition

$$\frac{k_f}{h}\frac{dT_f}{dy} = -k_s\frac{dT_s}{dy}.$$ (18)

A uniform temperature, $T_i$, is applied at the film inlet which is calculated assuming mass and heat continuity with the outlet and oil supply

$$T_i = \frac{V_o T_o + V_s T_s}{V_i}$$ (19)

where $V_o$ and $V_s$ are the volume flow rate from the film outlet and oil supply, respectively, $T_s$ is the temperature of the supply fluid and $T_o$ is the average temperature of the fluid from the outlet. The oil supply flow rate, $V_s$, is calculated from the difference in $V_i$ and $V_o$.

A fixed gradient of 0 is applied to the temperature at the outlet of the film region. A fixed gradient of 0 is also applied at the sides of the film region. The free convection condition is applied on the external solid boundaries and at the groove region.

$$\dot{Q} = h_c(T - T_{amb}) \tag{20}$$

where $h_c$ is the convective coefficient.

### 2.4. Numerical Setup

#### 2.4.1. Mesh Setup

The film region is a hexahedral mesh where the domain is discretised in the axial, circumferential, and radial directions, as shown in Figure 2a. Two meshes are present in the film region: a quasi two-dimensional mesh with a single radial cell where Equation (14) is applied, and a three-dimensional mesh where the energy equation, Equation (21), is applied as it requires some radial discretisation to capture the cross-film variation. The axial and circumferential discretisation is equal for both meshes. The bush mesh is shown in Figure 2b where the boundary on the inner surface is conformal with the film mesh. The shaft region is also conformal with the film mesh.

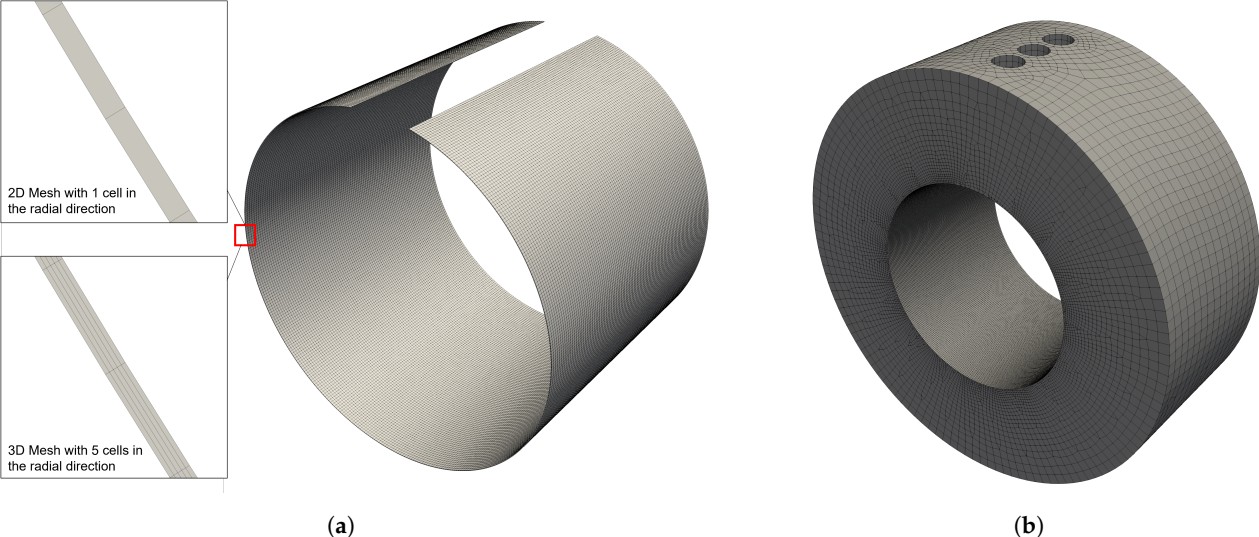

**Figure 2.** Images of the OpenFOAM mesh setups: (**a**) Film, (**b**) Bush.

The quasi-two-dimensional film mesh consists of 40,000 cells; 400 cells in the circumferential direction and 100 cells in the axial direction. A five-cell radial discretisation is applied in the three-dimensional film mesh for the energy equation, similar to the radial discretisation used in [4,25]. The bush mesh has 1,574,304 cells and the shaft mesh has 1,232,500 cells.

#### 2.4.2. Discretisation

The equations are discretised using the finite volume method. The cell thickness in the radial direction is set equal to the bearing clearance, $c$, while the film height is treated as a scalar field with a scalar value, h, for each cell. Implementing Equation (14) on this setup effectively reduces the equation to its 2D form inside a 3D CFD framework, while also allowing the film height to vary without changing the mesh on consecutive iterations.

The Laplacian term in Equation (14) is discretised with a central-difference scheme. In the full-film region, the divergence term in Equation (14) is also discretised with a central-difference scheme as the equation is in an elliptic form. In the cavitation region, the Laplacian term is removed by the switch function and the equation becomes hyperbolic. The divergence term is discretised using an upwind scheme in the cavitation region to

improve stability. The deformation of the surfaces is calculated explicitly from the pressure field using a midpoint numerical integration of Equation (4).

The non-dimensional film height, $\bar{h}$, is included in Equation (7) to account for the mesh setup.

$$\nabla \cdot (c_p \rho \bar{h} T) - \nabla \cdot \left( \frac{k}{\bar{h}} \nabla T \right) = \mu \left( \frac{U}{h} \right)^2 \qquad (21)$$

The convective term in Equation (21) is discretised using an upwind scheme and the Laplacian term is discretised using a central-difference scheme.

### 2.4.3. Procedure

The domain is discretised into individual regions, which are categorised as TEHL, bush and shaft. The regions are solved sequentially, as seen in Figure 3, in an iterative procedure until the convergence criterion is reached. The relative error between the temperature fields on subsequent iterations is used as the convergence criterion where a tolerance of $1 \times 10^{-5}$ is applied.

An iterative procedure is used in the TEHL region to converge to an input load. An initial estimate for eccentricity is used to initialise the film thickness field using Equation (3) with no elastic deformation of the surfaces. The switch function, g, is uniformly set as 1 across the domain, assuming no cavitation at initialisation. The temperature field is initialised uniformly as the supply temperature and the viscosity is subsequently calculated from Equation (9). The density is initialised as the fluid density while the remaining fluid property fields are initialised using Equation (15). Due to the integral in Equation (4), the elastic deformation function has a complexity of $O(n^2)$ where n is the number of cells in the two-dimensional mesh. To reduce the impact on the computation time, the cavitation algorithm is arranged in a sub-loop, as seen in Figure 3, to minimise the iterations of the deformation function required as the cavitation algorithm requires a large number of iterations to reach convergence. The cavitation algorithm includes the modifications described in Fesanghary et al. [11]. The convergence of the pressure within the cavitation algorithm is monitored using the relative error between iterations of the cavitation algorithm. Following convergence of the cavitation algorithm, the film thickness, fluid properties and temperature are calculated and convergence is tested based on the relative error of the pressure and temperature.

A stabilised preconditioned bi-conjugate gradient linear solver with a diagonal incomplete LU preconditioner is used for $\Theta$ and T in the film region. A generalised geometric-algebraic multi-grid linear solver with a Gauss Seidel smoother is used for the temperature solution in the bush and shaft regions. To reduce oscillations and accelerate convergence in the film region, the temperature and film thickness fields use explicit under-relaxation with factors of 0.8 and 0.6, respectively. The convective terms in the temperature equations for the film and shaft regions are discretised with a first-order upwind scheme. In the film region, the Couette term in the Equation (14) is discretised with a central difference scheme in the full-film region and an upwind scheme in the cavitation region. The Laplacian terms of the equations in all regions are discretised with a central difference scheme. The simulations are run on an Intel Xeon 3.9 GHz processor across eight cores. A tolerance value of $1 \times 10^{-5}$ is used for the convergence criterion of the field in the TEHL and solid regions. The load is considered converged when it is within 1% of the target load.

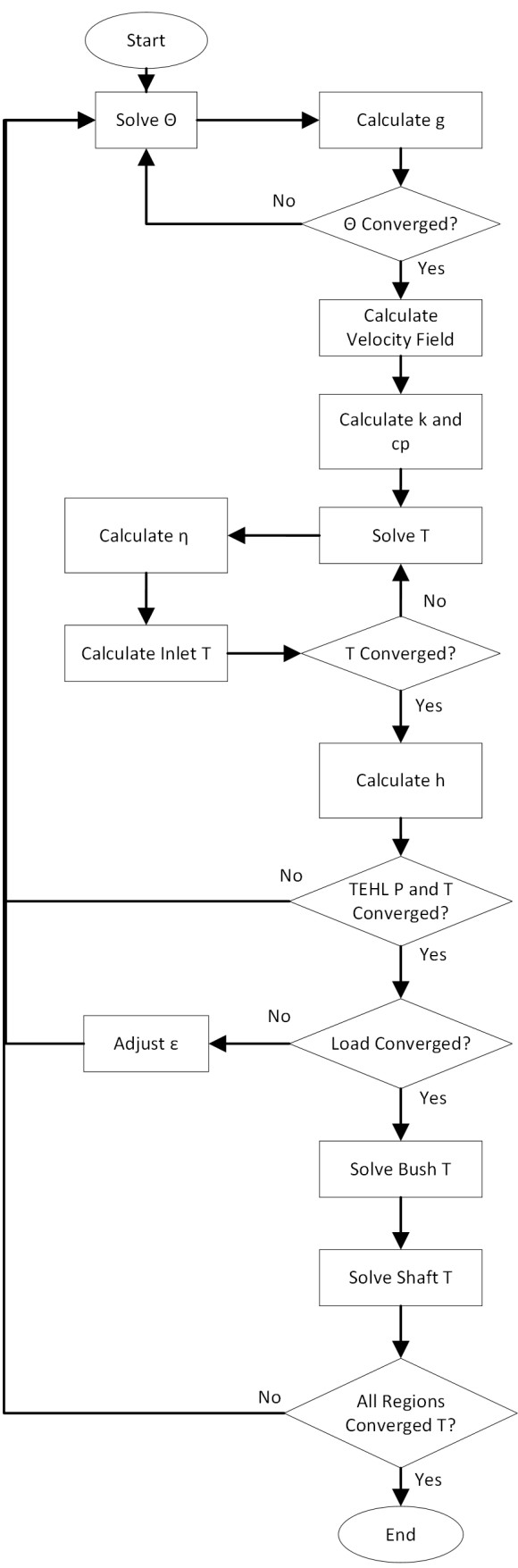

**Figure 3.** Flowchart of the solver procedure.

## 3. Results and Discussion

### 3.1. Case Description

The validation case is taken from Ferron et al. [26] of a single axial groove journal bearing. The OpenFOAM model is shown in Figure 1 with the bearing dimensions given in Table 2. The material and lubricant properties are given in Table 3. Gaseous cavitation is assumed where the vapor phase is comprised of undissolved air from the lubricant, which informs the material properties in the cavitation region. The bush and shaft materials are bronze and steel, respectively. The uncertainty of the thermal conductivity of the bush was stated in Ferron et al. [26], and lower values had be used in Banwait and Chandrawat [16] and in Stefani et al. [15].

**Table 2.** Bearing Dimensions [26].

| Property | Symbol | Value |
|---|---|---|
| Shaft Radius (mm) | $R_s$ | 25 |
| Bush Radius (mm) | $R_s$ | 50 |
| Bush Length (mm) | $L$ | 80 |
| Clearance (μm) | $c$ | 145 |
| Groove Angle (°) | $\theta_g$ | 18 |
| Groove Length (mm) | $L_g$ | 70 |
| Feed Hole Diameter (mm) | $D_g$ | 14 |
| No. Feed Holes | | 3 |

**Table 3.** Material and Lubricant Properties [26].

| Symbol | Value | Units |
|---|---|---|
| $P_c$ | 0 | Pa |
| $\beta$ | $1 \times 10^7$ | Pa |
| $\rho_f$ | 860 | $\frac{\text{kg}}{\text{m}^3}$ |
| $\rho_g$ | 1.225 | $\frac{\text{kg}}{\text{m}^3}$ |
| $c_{p_f}$ | 2000 | $\frac{\text{J}}{\text{kgK}}$ |
| $c_{p_g}$ | 1000 | $\frac{\text{J}}{\text{kgK}}$ |
| $\mu_f$ | 0.0293 | Pas |
| $\mu_g$ | $1.81 \times 10^{-5}$ | Pas |
| $k_f$ | 0.13 | $\frac{\text{W}}{\text{mK}}$ |
| $k_g$ | 0.05 | $\frac{\text{W}}{\text{mK}}$ |
| $\gamma$ | $-0.04$ | |
| $T_{ref}$ | 40 | °C |
| $k_s$ | 50 | $\frac{\text{W}}{\text{mK}}$ |
| $k_b$ | 100 | $\frac{\text{W}}{\text{mK}}$ |
| $\alpha_s$ | $1.2 \times 10^{-5}$ | $\text{K}^{-1}$ |
| $\alpha_b$ | $1.8 \times 10^{-5}$ | $\text{K}^{-1}$ |
| $E_s$ | 200 | GPa |
| $E_b$ | 113 | GPa |
| $v_s$ | 0.28 | |
| $v_b$ | 0.35 | |
| $h_{walls}$ | 50 | $\frac{\text{W}}{\text{m}^2}$ |
| $h_{groove}$ | 1500 | $\frac{\text{W}}{\text{m}^2}$ |

### 3.2. Mesh Study

The radial discretisation of the three-dimensional mesh in the film region, where Equation (21) is applied, affects the maximum bush temperature due to the gradients predicted at the film–solid interfaces.

A mesh sensitivity study was performed at a rotational speed of 4000 rpm and 6 kN load. Figure 4 shows the effect of varying the number of cells in the radial direction for the energy equation between 1–10. Independence is shown as the profile of the maximum temperature across the bush surface approaches 60 °C. At four cells in the radial direction, the error from the previous discretisation is <1% and is considered independent at this level.

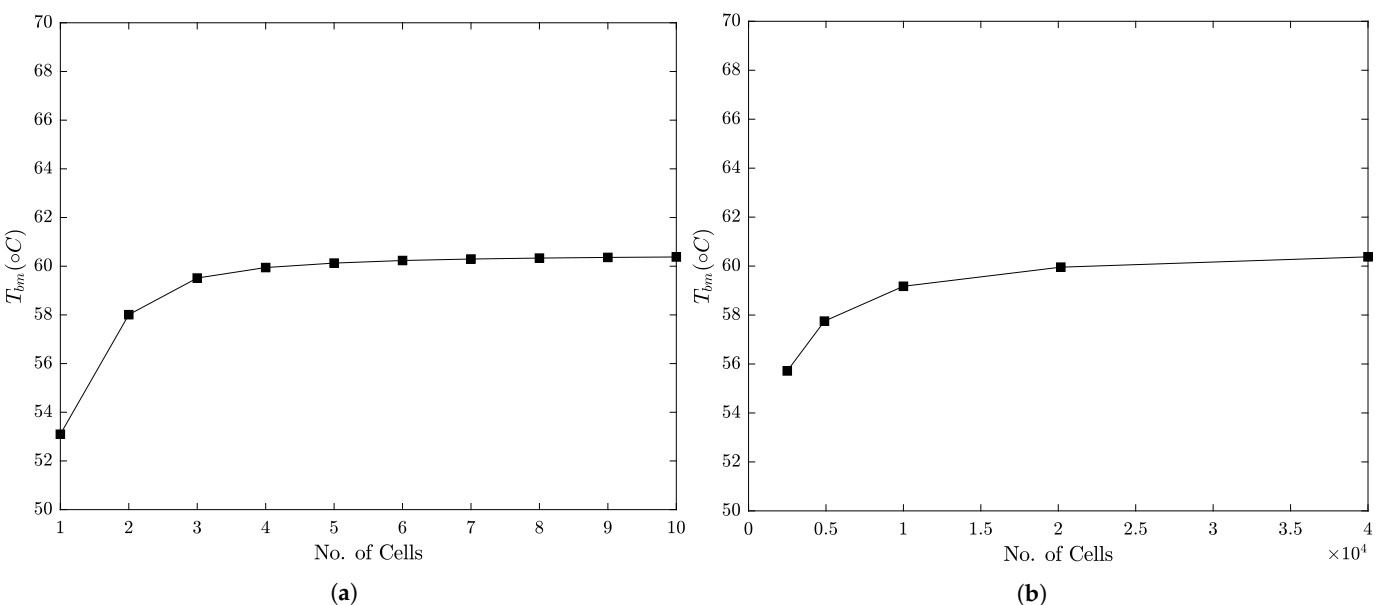

(**a**)    (**b**)

**Figure 4.** Mesh independence study: (**a**) Radial discretisation, (**b**) Surface discretisation.

Figure 4b shows the effect of varying the discretisation of the cells in the circumferential and axial directions with a constant discretisation in the radial direction of eight cells. The x-axis corresponds to the number of cells in the two-dimensional mesh which are doubled on each consecutive case. The solution is shown to approach mesh independence where the relative error from doubling the mesh size is <1% at 40,000 cells.

*3.3. Validation*

A parametric sweep was performed across a rotational speed range of 1500–4000 rpm and a load range of 2–10 kN, matching the condition in Ferron et al. [26]. The average computational time for a case was 54 min and the the maximum computational time was 123 min. Comparatively, the maximum computation time for the isothermal CFD simulation in Dhande et al. [27] was 180 min. This demonstrates the reduction in computational load given the lower computational time while including thermal effects.

A comparison of the centerline pressure in the film is shown for two cases: case a at 2000 rpm and 4 kN in Figure 5a, and case b at 4000 rpm and 6 kN in Figure 5b. The profiles show good agreement with the experimental results from Ferron et al. [26] where the gradients are near identical, increasing to the peak pressure and descending to the minimum film thickness. The peak pressure is located similar to the experimental results in both cases at 200°, which also show agreement in the location of the cavitation boundary at 268°. Furthermore, the similarity between the peak pressure and the cavitation boundary locations supports the angular offset prediction in the OpenFOAM solution. The peak pressure tends to be underestimated due to the reduced bulk modulus values used in the cavitation model, as seen in the centerline profiles, where the peak value is underestimated by 0.1 MPa in case a and 0.22 MPa in case b. The mean error between the numerical and experimental pressure is 0.05 MPa in case a and 0.072 MPa in case b.

The centerline temperature profiles for the film–bush interface are shown in Figure 6. The trend in the profiles shows good agreement with the experimental results from Ferron et al. [26], with close similarity in the peak temperature location in both case a and b. The

peak temperature predicted by the OpenFOAM model is higher in both cases: 1 °C for case a and 2.5 °C for case b. Overestimation of the temperature is greater in the converging region compared with the peak value for both cases where, at the bush–groove interface, the OpenFOAM results are 2 °C and 4 °C higher in case a and b, respectively. This may suggest that the inlet temperature calculated from Equation (19) is overestimated in the film region, and that the cooling effect of the groove region is not well accounted for in the numerical model. Integrating the temperature profiles along the centerline gives a representative value for the heat generated and dissipated to the solid region, which can be compared with the experimental results,

$$P_d = \int T * R d\theta \tag{22}$$

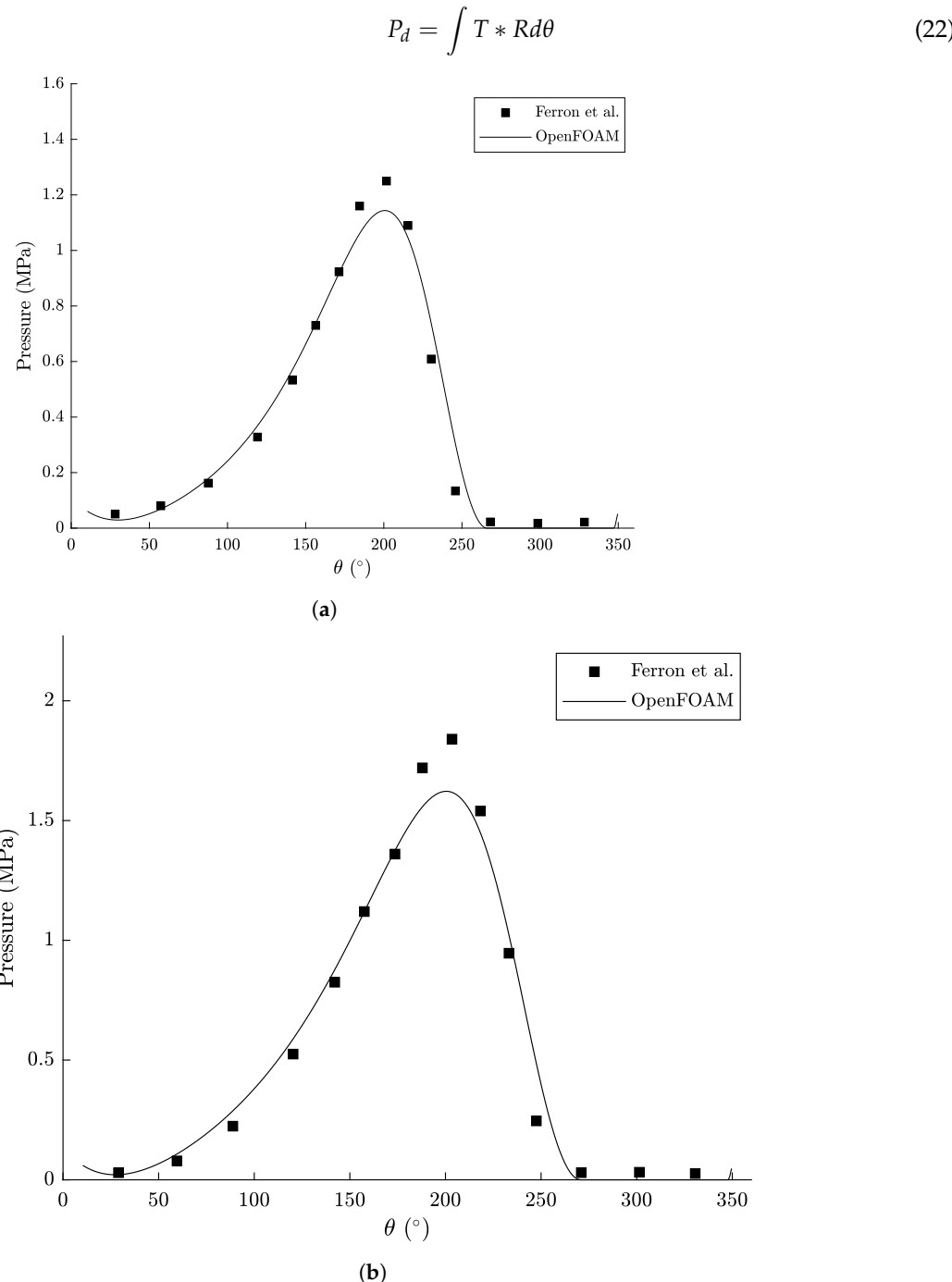

**Figure 5.** Film centerline pressure profiles compared with Ferron et al. [26]: (**a**) 2000 rpm and 4 kN, (**b**) 4000 rpm and 6 kN.

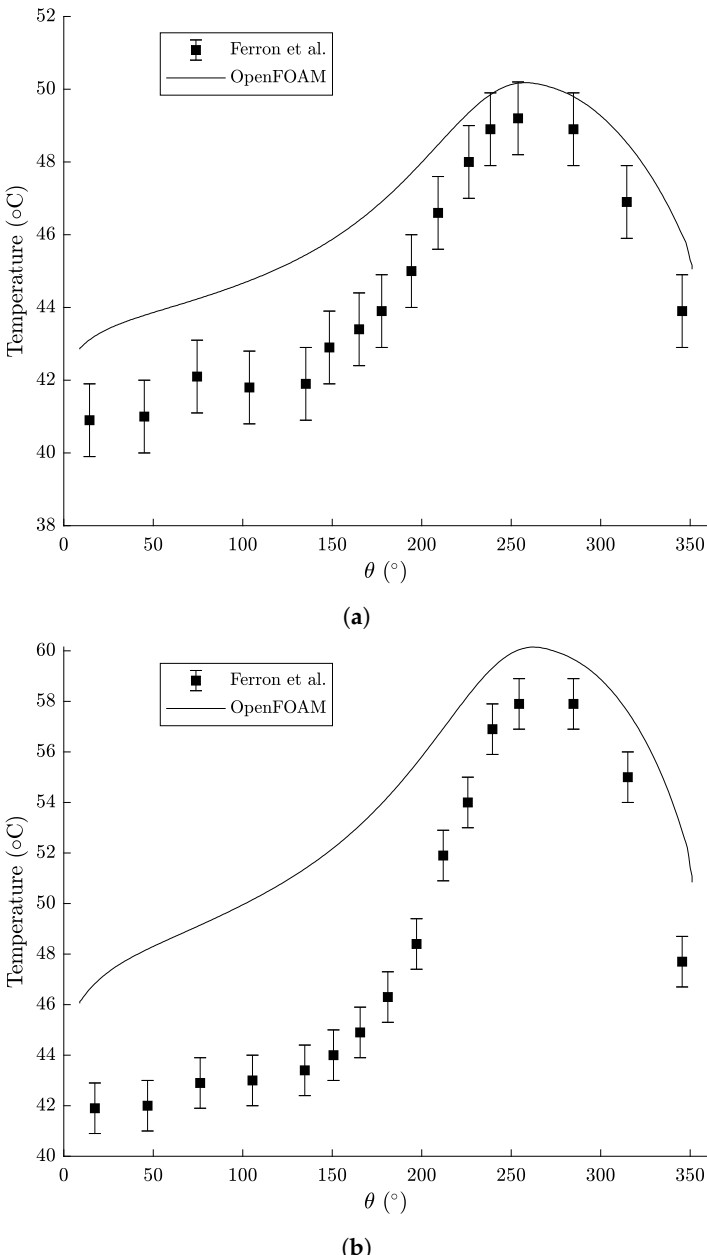

**Figure 6.** Bush–film interface centerline temperature profiles compared with Ferron et al. [26]: (**a**) 2000 rpm and 4 kN, (**b**) 4000 rpm and 6 kN.

Using a trapezoidal integration on case a gives a difference of 8.3% between the numerical and experimental results, and 15.8% in case b. Only the viscosity is considered as a function of temperature while the thermal properties are considered constant for the fluid phase and in the solid regions. The various fluid properties in the film will vary with temperature, including the thermal conductivity, specific heat capacity and density, which the current model does not consider. Variation in these properties could influence the temperature solution at higher operating speeds and loads where there is a significant change in temperature from the ambient conditions and where the temperature variation within the solution is greater. Furthermore, the temperature variation in the solid region will also vary the thermal conductivity of the bush, which is not considered in the numerical model.

The temperature of the fluid at the film inlet is calculated by applying the energy and mass continuity between the outlet and the inlet using Equation (19), therefore, approximating the effect of mixing in the groove without modelling the interaction with the oil

supply. Furthermore, free convection is assumed at the bush–groove interface; therefore, energy continuity is not applied with the bush region. These factors could affect the film inlet temperature and the temperature solution in the bush given the convection coefficient is estimated from previous studies, causing the difference in temperature results seen in Figure 6. Applying a CFD model to simulate the mixing in the groove region could improve the inlet temperature of the film and the solution in the bush by coupling across the bush–groove interface. To consider the results without the groove mixing, the center-line temperature profile in Figure 6 is normalised to the temperature at the bush–groove interface, as presented in Figure 7. The normalised results highlight the agreement in the temperature gain within the film, where a higher gradient is seen in the converging region and a lower difference between the inlet and maximum temperature. Integrating the normalised temperature using Equation (22) gives a difference between the numerical and experimental results of 3.3% in case a and 4.9% in case b. This shows that greater agreement in the heat generation is seen when the temperature is normalised to the inlet temperature.

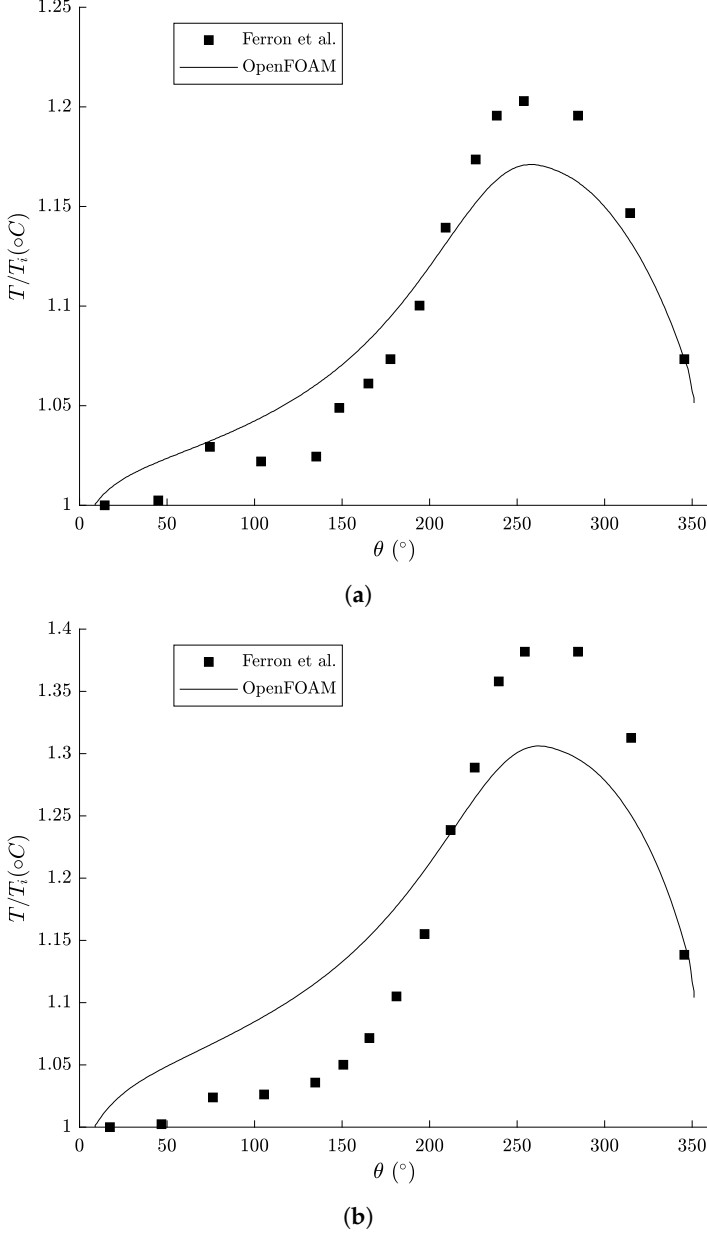

**Figure 7.** Bush–film interface centerline temperature profiles normalised to temperature at the groove compared with Ferron et al. [26]: (**a**) 2000 rpm and 4 kN, (**b**) 4000 rpm and 6 kN.

The relation between eccentricity and load is presented in Figure 8 compared against the results from Ferron et al. [26]. The experimental eccentricity is calculated using the clearance in the original study and some uncertainty is noted in these values due to the thermal expansion of the solid components during operation. An adjusted clearance value was calculated by assuming a higher temperature in the shaft and bush and was used to calculate the corrected data set which was shown to be closer to the theoretical values. Similarly, the eccentricity from the OpenFOAM results showed more agreement with the corrected experimental data, particularly in the profile where there was a highly uniform difference between the eccentricity values—approximately 0.1 difference. Given the uncertainty from the methodology for calculating the eccentricity in the experimental case and the uniform difference in the eccentricity values, the experimental results are adjusted in the following figures by reducing the eccentricity by 0.1.

A comparison of the peak pressure in the film against the shaft eccentricity is shown in Figure 9. With the adjustments to the experimental eccentricity, the results show good agreement in trend and magnitude. The results show the peak pressure is predicted generally within 0.5 MPa where, at higher eccentricity, corresponding to higher load, the peak pressure begins to be underestimated. Similarly, the peak temperature values for the bush–film interface are compared with the experimental results in Figure 10. At low rotational speed, the maximum bush temperature prediction shows good agreement with the experimental results. The magnitude is within 8% across the parametric study, <5% at the lower rotational speeds. A higher speed and low eccentricity shows the largest difference between the experimental and numerical results.

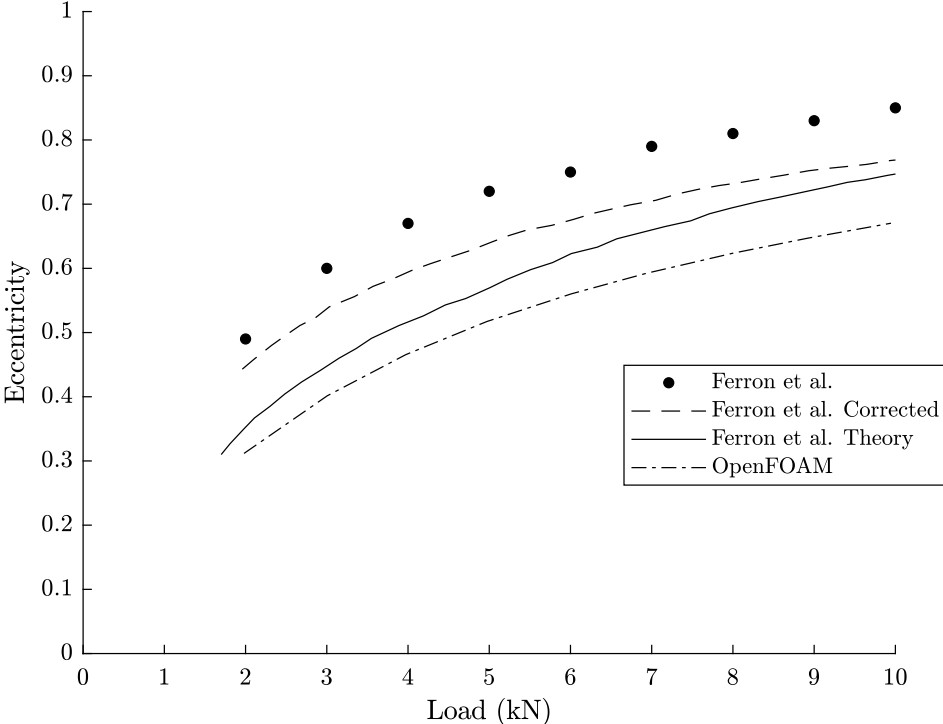

**Figure 8.** Eccentricity against load At 2000 rpm compared with Ferron et al. [26].

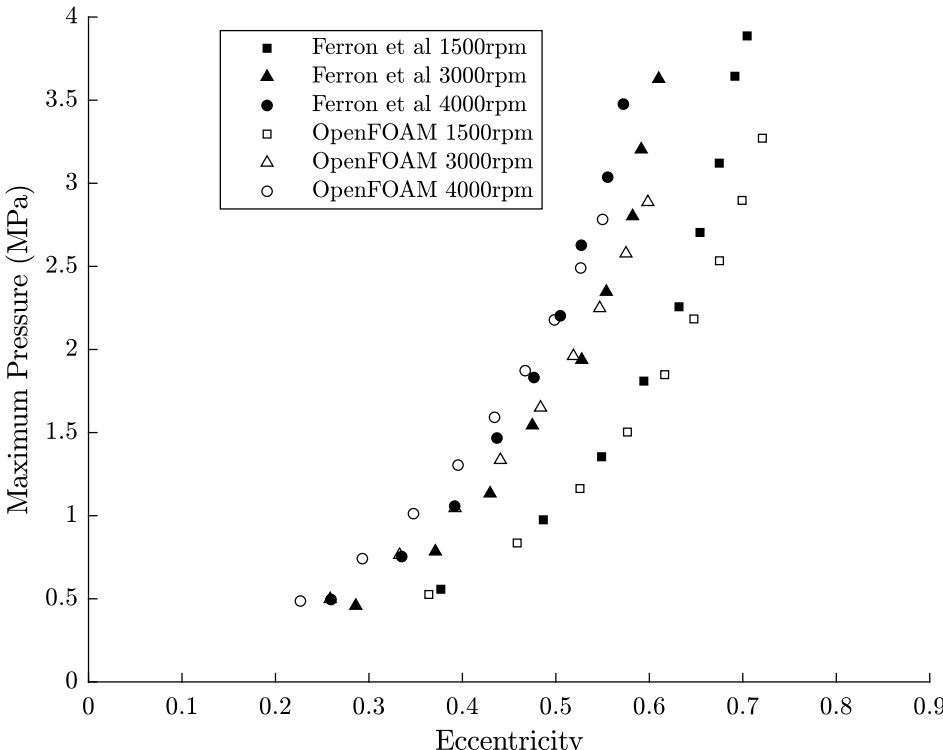

**Figure 9.** Peak pressure in the film. compared with Ferron et al. [26].

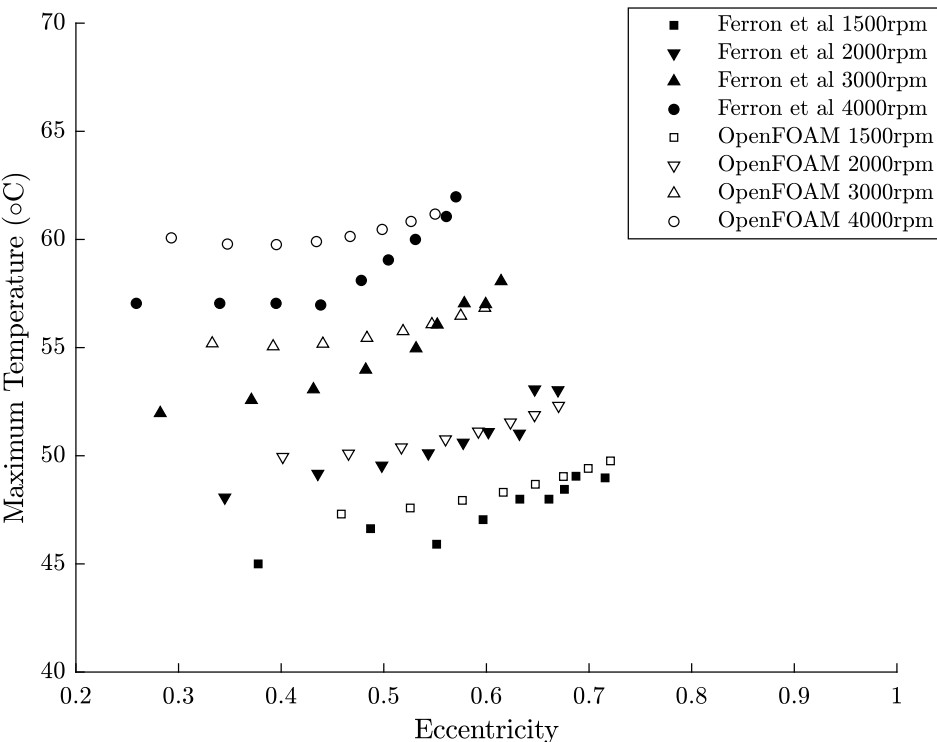

**Figure 10.** Peak temperature on the film–bush interface compared with Ferron et al. [26].

The contours of pressure are shown in Figure 11 to present the full profile within the film where the flow is in the positive $\theta$ direction. The pattern of peak pressure can be seen, whereby, as the load increases and the speed decreases, the maximum pressure increases. The results also show the trend in the pressure profile to spread as the rotational speed is increased and the peak pressure decreases.

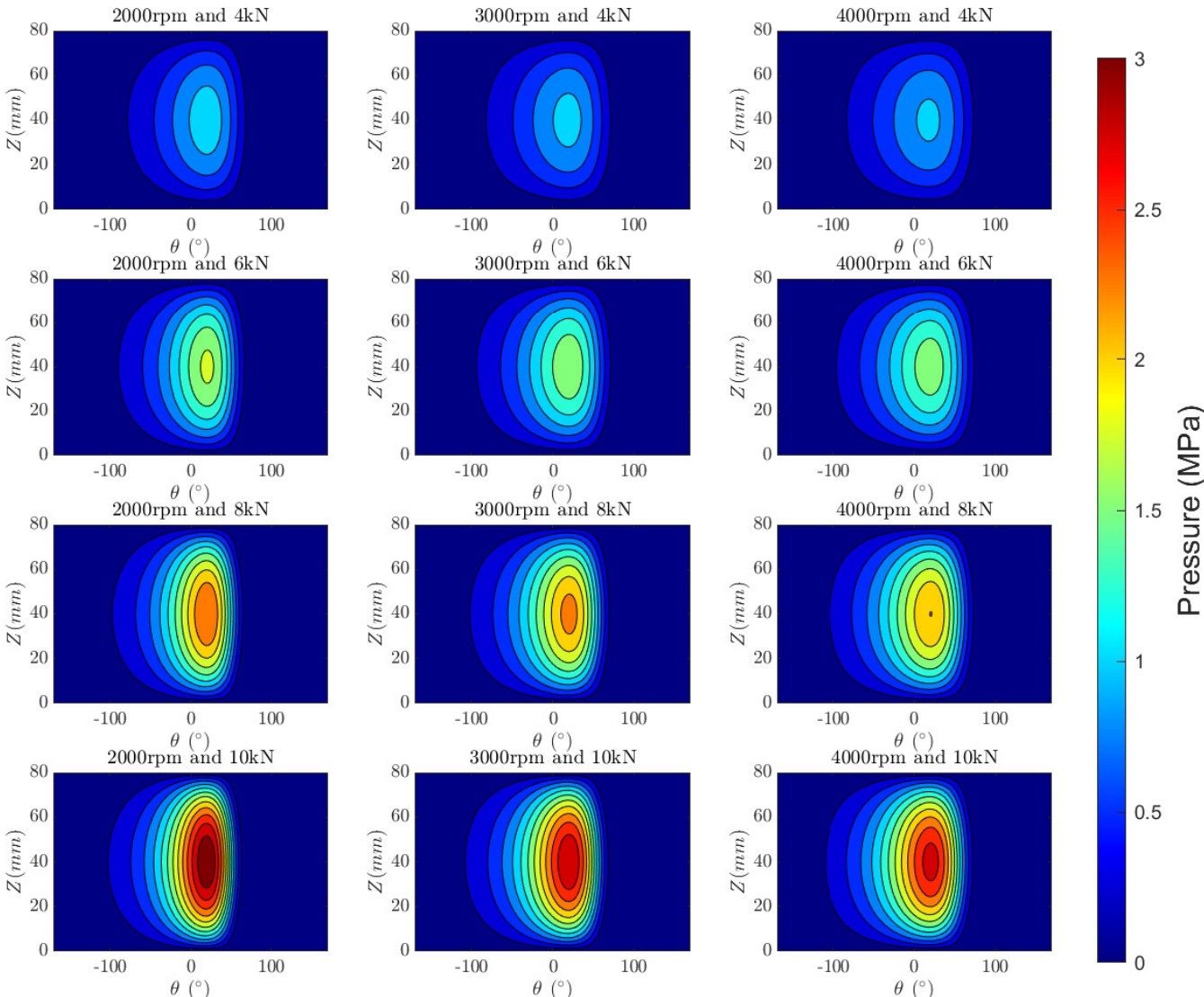

**Figure 11.** Contours of film pressure for 2000–4000 rpm rotational speed and 4–10 kN load.

Similarly, the temperature profile across the bush surface is shown in Figure 12 where the axial variation in temperature can be seen. The temperature variation in the film is shown in Figure 13 along the centerline of the bearing. The same trend in temperature is seen, whereby, increasing the load and the rotational speed increases the maximum temperature. The temperature variation across the film height can be seen and compared.

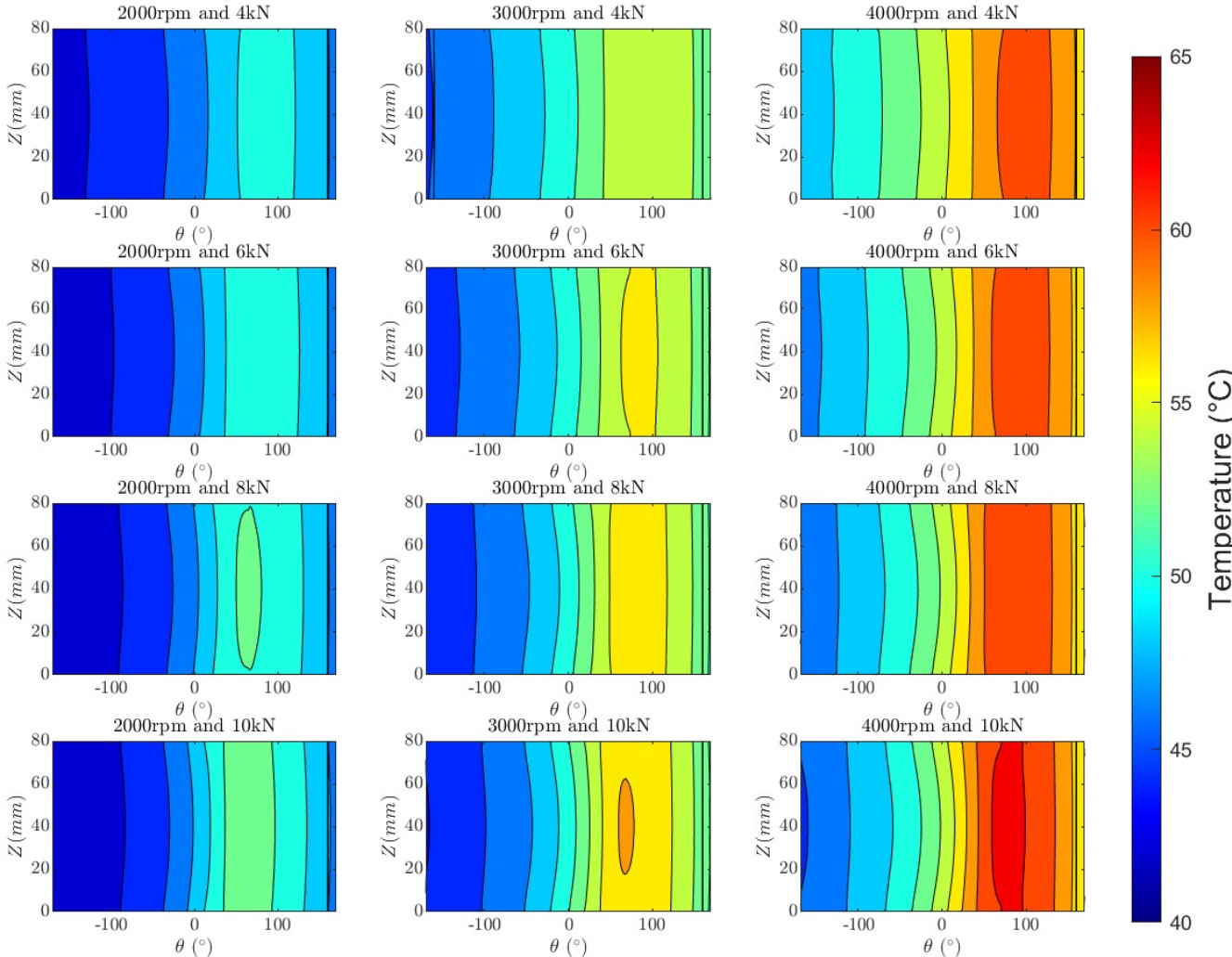

**Figure 12.** Bush–film temperature contours.

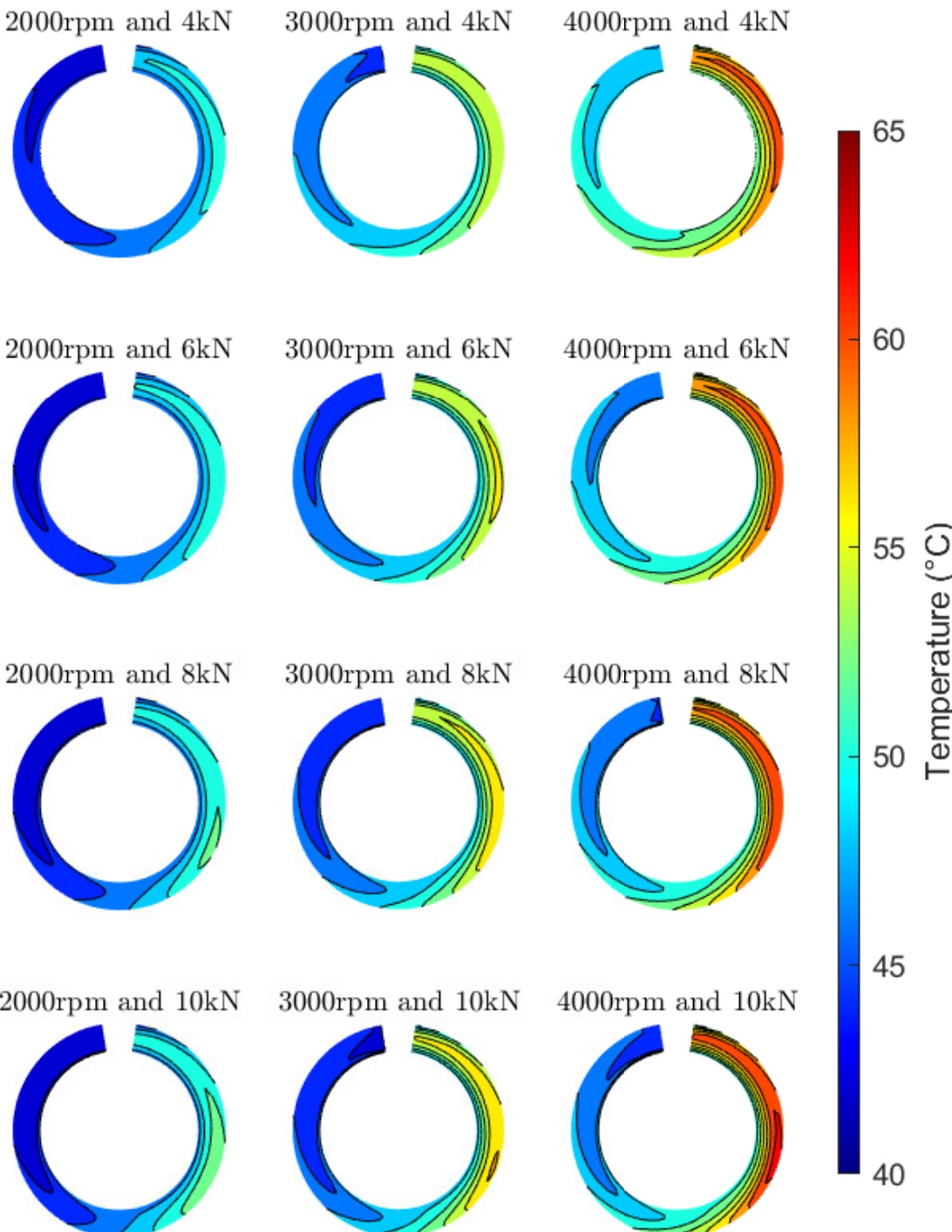

**Figure 13.** Film centerline temperature contours.

## 4. Conclusions

This research paper presents a comprehensive investigation into a Reynolds-based hydrodynamic lubrication model, which incorporates crucial factors, such as cavitation, thermal effects, and elastic deformation, and its accuracy is validated against experimental data from a single axial groove journal bearing.The results demonstrate the effectiveness of the methodology. The pressure values deviate by an average of approximately 5% from the experimental values and the peak pressure is estimated within 1 MPa. Similarly, a comparison of temperature on the bush surface reveals an average deviation of approximately 5%

from the experimental data, with the peak temperature predicted within approximately 2% across the experimental range. Notably, the largest discrepancy in the predicted temperature profiles occurs in the converging region where the numerical predictions fall outside the experimental error range. This suggests that the methodology is unable to capture mixing processes in the groove. Additionally the model was applied to the same bearing configuration and a parametric study conducted, highlighting the pattern of peak pressure in response to variations in the load and rotational speed. Increasing the load consistently increases the peak pressure, while increasing the rotational speed decreases it. Moreover, the study demonstrates that temperature in the lubricant film rises with both rotational speed and load.

The implementation of this methodology using the OpenFOAM source code provides a foundation for further development and integration with CFD models in regions beyond the hydrodynamic lubrication regimes, such as investigating the mixture of oil in the gaps within the tilting pad bearings. Applying a CFD methodology within the groove region could enhance temperature prediction around the lubricant supply, while extending the simulation to a multiphase CFD model of the wider domain could improve the outer thermal boundaries on the solid, eliminating the need to estimate the convection coefficient. Further avenues for future work could include incorporating alternative cavitation methodologies based on fluid pressure-density models instead of switch functions. Additionally, the inclusion of temperature-dependent material properties, such as the thermal conductivity and specific heat capacity, as well as fluid properties, such as the density and pressure-dependent viscosity. that would be relevant at higher loads, could enhance the accuracy of the model.

**Author Contributions:** Conceptualization, J.L., B.C.R., S.A. and C.E.; formal analysis, J.L.; project administration, C.E.; software, J.L.; supervision, B.C.R., S.A., C.E. and H.M.; Visualization, J.L.; writing—original draft, J.L.; writing—review and editing, B.C.R., S.A., H.M. and N.R. All authors have read and agreed to the published version of the manuscript.

**Funding:** This work was funded by the Prosperity Partnership Grant Cornerstone: Mechanical Engineering Science to Enable Aero Propulsion Futures, Grant Ref: EP/R004951/1.

**Data Availability Statement:** The data that support the findings of this study are available on request from the corresponding author, Benjamin C. Rothwell.

**Acknowledgments:** The authors thank Rolls-Royce plc and the EPSRC for the support under the Prosperity Partnership Grant Cornerstone: Mechanical Engineering Science to Enable Aero Propulsion Futures, Grant Ref: EP/R004951/1.

**Conflicts of Interest:** The authors declare no conflict of interest.

## Nomenclature

| | |
|---|---|
| $\alpha$ | Thermal expansion coefficient |
| $\zeta$ | Lubricant properties ($\mu, c_p, k$) |
| $P$ | Lubricant pressure (Pa) |
| $P_0$ | Reference pressure (Pa) |
| $h$ | Film height (m) |
| $h_e$ | Film height generated from the eccentricity of the shaft (m) |
| $h_d$ | Surface deformation (m) |
| $h_T$ | Thermal dilation (m) |
| $T_f$ | Temperature in the lubricant film (°C) |
| $T_b$ | Temperature in the bush (°C) |
| $T_{ref}$ | Reference temperature (°C) |
| $\Theta$ | Relative density |
| $g$ | Switch variable |

| | |
|---|---|
| $\rho$ | Lubricant density ($\frac{\text{kg}}{\text{m}^3}$) |
| $\rho_0$ | Lubricant density at reference pressure (kg/m$^3$) |
| $\mu$ | Dynamic viscosity (*Pas*) |
| $\mu_0$ | Dynamic viscosity at reference temperature (Pas) |
| $U_s$ | Shaft surface velocity ($\frac{\text{m}}{\text{s}}$) |
| $\vec{U}_c$ | Couette term of velocity ($\frac{\text{m}}{\text{s}}$) |
| $x, y, z$ | Coordinates in the circumferential, radial and axial directions, respectively |
| $c$ | Clearance (m) |
| $\epsilon$ | Eccentricity |
| $\theta$ | Angular coordinate ($\circ$) |
| $e$ | Distance between shaft and bush axes (m) |
| $E$ | Young's modulus (Pa) |
| $v$ | Poisson's ratio |
| $c_p$ | Lubricant specific heat capacity ($\frac{\text{J}}{\text{kgK}}$) |
| $k_f, k_g$ | Thermal conductivity of the lubricant liquid and gaseous phases, respectively ($\frac{\text{W}}{\text{mK}}$) |
| $k_s, k_b$ | Thermal conductivity of the shaft and bush, respectively ($\frac{\text{W}}{\text{mK}}$) |
| $\hat{x}$ | Unit vector parallel to the bush in the direction of the flow |
| $\bar{h}$ | Non-dimensional film height ($\frac{h}{c}$) |
| $P_d$ | Heat generation (Km) |
| $E$ | Young's modulus Pa |
| $v$ | Poisson's ratio |
| $R$ | Radius |

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
