# Peer review of "A New Thermal Elasto-Hydrodynamic Lubrication Solver Implementation in OpenFOAM"

_lubricants, doi:10.3390/lubricants11070308_

Round 1

Reviewer 1 Report

I have provided extensive comments in the attached pdf. Please contact the journal if these are not clear and require further information to resolve.

There is novelty and the work is of interest to the readers of Lubricants. Particular attention needs to be paid in linking the work developed in the paper with visual aids. There are questions over the use of gradient operators within 2D and 3D problems which must be resolved. There is a lack of depth in the review of literature related to the research presented. The conclusions are weak and do not link to the wider literature, the results do not push the model to exploration. There was no future work recommended.

Critically, as the research has been produced using an open source software. This code should be made publicly available and referenced within the paper, the solver and cases should be provided to allow readers to recreate the work presented. This is also in line with the GNU agreement allowing the researchers to use the software. A github link to this or similar should be provided before publication can be considered.

There are several formatting errors in terms being used multiple times, the formatting of terms in the text, spacing errors throughout. Some small spelling errors, revise according to the pdf provided.

Reviewer 2 Report

1)      Some specific results have to be added in the Abstract.

2)      In the literature there are only 2 references from the last 5 years?

3)      State the objectives of the paper at the end of the Introduction.

4)      Line 200: ”where h is the convective”? – This is an incomplete sentence.

5)      Line 144: ”where cp is the specific heat capasity (fracJkgK)” – I suggest that authors remove the units from the main text, since you have Nomenclature at the end. In the Nomenclature, check/revise the units again. For example, revise the unit for specific heat capacity - Instead of J/kgK write J/(kgK)… see also thermal conductivity.

6)      Table 2. Unit  K-1. Put subscribe.

7)      Line 204. …2 dimensional 204 mesh as shown in Figure 2…. – There is no mesh in Fig. 2!?

8)      Expand the Conclusion with the main results and implications of the study.

Minor editing of English language required

Reviewer 3 Report

Dear Gentlemen,

Thank you very much for the interesting article. I see potential in your approach. Since it is the first step according to your words, the scope and function is completely fine. However, there are many TEHL-tools with sometimes much wider range of functions. Therefore, I miss in some places a justification why some features were not or can not be modelled, as well as the explicit advantage of your software compared to other tools (e.g. no quantitative time advantage is given). I therefore hope that the following comments will help you to:

Line 8: See comment no 4

Line 11: Mixing is not clear at this point. Pleace specify that. 

Line 17: Formatting error

Line 37 following: There is a wide range on TEHL-simulation in state of knowledge. The calculation time is depending on the simpilfication used, the calculation method (FEM, halfspace, ...), the solving strategy, ... . The general statement TEHL simulations are more time efficient than CFDs is therefore wrong. A more precise differentiation must be made here.

Page 4: Several formatting errors

Line 150: Which lubricant is used and how are the parameters derived? Pleace add the missing information. 

Why is the pressure dependency neglegted? Pleace justify this simplification.

Why there is no temperature and/or pressure dependent density definition? Pleace justify this simplification.

Line 165: Please provide a source that this equation is allowed to be used here for the mentioned parameters.

Line 165: Formatting error

Page 6:  Several formatting errors

Line 221: It is not obvious which boundary condition is set where. Please add a schematic illustration with the labels of the boundaries and the domains.

Line 270: Operating conditions in rotational speed and normal force are uncommon for TEHL simulations. Please also give the load with the Hertzian pressure and the rotational speed with the mean velocity and the slide to roll ratio.

Line 286: Since the time advantage is the rationale for this model, pleace give information on the calculation time and the time advantage.

Line 299: "Good" is to optimistic with 5 °C temperature difference at a level of ~45 °C. 

Line 317: There are no information, which of the mentioned points were considered by Ferron et al. Pleace add these information.

I hope these comments help you with your article. I look forward to reading the finished article.

Round 2

Reviewer 1 Report

All of my comments have been sufficiently responded to given the circumstances and I am in agreement that the paper should be published in the current format.

Author Response

The authors would like to thank the reviewer for their assistance on this manuscript.

Reviewer 2 Report

The authors improved their paper. Only in   Table 2 Note Unit (K-1). Put in subscribe -1.

Author Response

Thankyou for the comment on Table 2, the formatting error has been amended.

Reviewer 3 Report

Dear Ladies and Gentlemen,

Thank you very much for the changes and the explanatory comments. From my point of view, there are no points that must be corrected.

Thank you very much for your interesting paper.

Author Response

(The authors gave the same response as above.)
